# Prediction of Metastasis in Paragangliomas and Pheochromocytomas Using Machine Learning Models: Explainability Challenges

**DOI:** 10.3390/s25134184

**Published:** 2025-07-04

**Authors:** Carmen García-Barceló, David Gil, David Tomás, David Bernabeu

**Affiliations:** University Institute for Computer Research, University of Alicante, Carretera San Vicente del Raspeig s/n, 03690 San Vicente del Raspeig, Spain; david.gil@ua.es (D.G.); dtomas@dlsi.ua.es (D.T.); davidbernabeuferrer@gmail.com (D.B.)

**Keywords:** machine learning, explainability, data science, classification, feature selection, tumor, metastasis

## Abstract

One of the main issues with paragangliomas and pheochromocytomas is that these tumors have up to a 20% rate of metastatic disease, which cannot be reliably predicted. While machine learning models hold great promise for enhancing predictive accuracy, their often opaque nature limits trust and adoption in critical fields such as healthcare. Understanding the factors driving predictions is essential not only for validating their reliability but also for enabling their integration into clinical decision-making. In this paper, we propose an architecture that combines data mining, machine learning, and explainability techniques to improve predictions of metastatic disease in these types of cancer and enhance trust in the models. A wide variety of algorithms have been applied for the development of predictive models, with a focus on interpreting their outputs to support clinical insights. Our methodology involves a comprehensive preprocessing phase to prepare the data, followed by the application of classification algorithms. Explainability techniques were integrated to provide insights into the key factors driving predictions. Additionally, a feature selection process was performed to identify the most influential variables and explore how their inclusion affects model performance. The best-performing algorithm, Random Forest, achieved an accuracy of 96.3%, precision of 96.5%, and AUC of 0.963, among other metrics, combining strong predictive capability with explainability that fosters trust in clinical applications.

## 1. Introduction

Currently, tumors represent one of the main concerns in the medical field due to their impact on human health, as well as their variability in origin, behavior, and response to treatment.

Metastasis [1] refers to the process by which cancer cells spread from the original location where they emerged to other parts of the body, typically through the bloodstream or lymphatic system, leading to the formation of secondary tumors. Importantly, the new tumor is of the same cancer type as the original one. This phenomenon is critical in cancer progression, being responsible for nearly 90% of deaths from undetected cancers [2].

The present study aims to analyze data associated with paragangliomas and pheochromocytomas (PPGLs), which are uncommon neuroendocrine tumors. Pheochromocytomas typically arise in the adrenal medulla, which is the inner part of the adrenal glands located on top of the kidneys, whereas paragangliomas occur outside the adrenal glands, often along nerve pathways in the head, neck, and spine [3]. These tumors can be either benign or malignant and have a notable hereditary component, with up to 35% of cases linked to genetic predisposition [4]. Metastatic PPGLs pose a significant clinical challenge because they are highly heterogeneous and can be potentially lethal [5]. In addition, these tumors can mimic the symptoms of more than 30 different medical conditions and have a low reported incidence [6]. Although the incidence has increased in recent decades, reaching approximately 0.6 cases per 100,000 people per year after 2010, PPGLs remain understudied and often underdiagnosed [7]. This highlights the need for improved diagnostic tools and a better understanding of their clinical behavior.

Advances in the field of medicine and technology have driven the development of more sophisticated tools for the detection and prediction of cancer, among which machine learning stands out as a promising discipline [8]. In this context, this type of tumor, represents an area of increasing interest due to its low incidence and diagnostic complexity. The early detection of metastasis is crucial for determining the course of treatment and improving the survival prognosis of patients.

While machine learning has shown significant potential for enhancing predictive accuracy in medicine, its adoption in critical fields such as oncology is often hindered by a lack of transparency. ML models are frequently considered black boxes and this opacity can undermine trust in their predictions. As these tools grow more advanced and complex, although their accuracy increases, it becomes harder to understand the rationale behind their decisions [9].

This challenge underscores the importance of explainability in machine learning, especially in high-stakes applications such as healthcare. Explainable Artificial Intelligence (XAI) techniques aim to bridge the gap between predictive performance and interpretability by providing insights into the decision-making processes of ML models [10]. To foster clinical adoption, these techniques are essential, as users need to understand how and why a model has arrived at a specific decision before making any clinical choices.

In light of these considerations, the main objective of this research is to develop a machine learning framework to predict the metastatic potential of PPGLs, with the aim of improving predictive performance over previous approaches while ensuring model interpretability through explainable AI techniques. To achieve this goal, this study adopts a procedure that integrates data analysis and preprocessing. In addition to evaluating model performance using the full feature set, a complementary analysis with a reduced set of selected features is performed to examine its impact on predictive capability. Furthermore, the study aims to improve classification accuracy by employing a variety of machine learning models, fine-tuning their hyperparameters, and evaluating their performance using appropriate metrics. Additionally, explainability techniques are employed to provide insights into the decision-making processes of these models, bridging the gap between accuracy and interpretability. This approach not only addresses the diagnostic challenges posed by PPGLs but also emphasizes the use of explainability techniques to build trust and transparency, facilitating the integration of machine learning tools into clinical practice.

The remainder of this paper is structured as follows: Section 2 presents an overview of related work on the use of machine learning and explainability techniques in medical applications; Section 3 defines explainability, reviewing its types and common techniques in machine learning; Section 4 describes the proposed methodology; Section 5 presents the results and evaluation of the models, including the application of explainability techniques; Finally, the paper concludes with Section 6, which outlines the main findings and proposes directions for future research.

## 2. Related Work

Recent developments in computer science have significantly impacted healthcare, particularly through the integration of machine learning. As a branch of artificial intelligence, machine learning leverages computational algorithms to analyze multidimensional data. In disease diagnosis, these algorithms are primarily used for classification tasks [11].

Previous studies have contributed to a better understanding of renal cancer progression. Jagga and Gupta [12] developed a series of supervised machine learning classification algorithms to enhance the analysis of disease progression by classifying tumor stages based on gene expression data obtained through RNA sequencing (RNA-seq) [13] experiments, a technique that quantifies RNA levels to reveal the activity of genes in a sample. Their study used data from The Cancer Genome Atlas (TCGA) dataset [14]. Specifically, renal tumors of the PPGL type have been the focus of efforts to develop scoring systems that combine multiple features for predicting metastasis. However, most of these approaches rely on histopathological parameters, such as tumor cell characteristics and tissue invasiveness [15,16], which are difficult to standardize in clinical practice [17] and often exhibit limited accuracy [18]. In an effort to improve metastatic risk prediction for PPGLs, in 2018, Cho et al. [19] developed the ASES score, a clinical model incorporating age, tumor size, tumor location, and secretory type to estimate metastatic potential. While this model provides a practical tool based on easily obtainable clinical parameters, its predictive accuracy remains moderate, highlighting the need for more robust, data-driven approaches, such as machine learning, to enhance metastasis prediction.

The present work is grounded in the study by Pamporaki et al. [4], who developed machine learning models to predict metastatic disease in PPGLs, aiming to achieve greater accuracy than previous approaches. They used supervised learning algorithms, including Decision Tree, Support Vector Machines (SVM), Naive Bayes, and an ensemble of decision trees with AdaBoost. The results showed that the most effective model, based on the ensemble approach, achieved an area under the ROC (Receiver Operating Characteristic) curve of 0.942. The balanced accuracy was 88%, with sensitivity of 83% and specificity of 92% in the external validation set. These findings underscore the potential of machine learning models in accurately predicting metastatic disease in PPGLs. More recently, Alzahrani and Alharithi [20] applied machine learning to enhance the detection of rare genetic disorders, including PPGLs, using whole-genome sequencing data from the Swedish genomics database (SweGen) [21]. They compared the performance and time cost of advanced techniques such as Artificial Neural Networks (ANN) and Random Forest with classical algorithms like SVM, Logistic Regression, and K-Nearest Neighbor (KNN).

Furthermore, determining reliable features for predicting the metastatic potential of PPGLs at the time of initial diagnosis is essential. Several clinical features have been suggested as potential indicators of metastatic risk in PPGLs. Some studies have reported that younger age [22,23,24], larger tumor size [22,23,24,25], extra-adrenal location of primary tumors [22,25], and the presence of mutations in the SDHB gene [26,27] are associated with a higher metastatic risk.

Recent research has explored the impact of explainability in the context of medical machine learning. In this context, Allgaier et al. [28] conducted a systematic review on the use of explainability methods in healthcare-related ML applications, finding that many models lack such techniques. Among those that do incorporate explainability, SHAP [29] and Grad-CAM [30], for imaging tasks, are the most commonly used. The study further highlights the importance of transparency in medical AI and the need to enhance model interpretability for clinicians and patients. No studies were found that specifically address explainability in renal cancer; however, notable progress has been made in other cancer types. For instance, Alabi et al. [31] applied machine learning techniques to predict survival in nasopharyngeal cancer, using LIME [32] and SHAP interpretability methods to assess the impact of various factors on predictions. Similarly, Jansen et al. [33] explored explainability in breast cancer survival prediction, and Pathan et al. [34] investigated its application in lung cancer. The field of machine learning explainability is advancing rapidly, with continuous research underscoring its increasing relevance in the context of clinical oncology.

## 3. Explainability Methods

In the context of machine learning, explainability refers to the ability of a model to provide clear and understandable insights that enable humans to understand the reasoning behind its predictions [35].

There are various criteria for classifying types of explainability. In this case, we will focus on the criterion that distinguishes between global and local techniques [36]. Global methods describe the overall behavior and effect of input features in machine learning models, while local methods focus on explaining predictions for a single instance, such as an individual patient. Typical examples of global approaches include Partial Dependence Plots (PDPs) [37] and Feature Importance [38]. On the other hand, widely used local methods include SHapley Additive exPlanations (SHAP), Local Interpretable Model-Agnostic Explanations (LIME), and Individual Conditional Expectation (ICE) [39]. Notably, SHAP can also be applied for global explanations.

Some machine learning models are inherently interpretable, meaning their structure and decision-making process are transparent by default. Examples of such models include Linear Regression, Logistic Regression, Naive Bayes, or KNN. These models provide clarity and can be considered “interpretable by nature” [40]. In contrast, more complex models, such as ensemble models (e.g., Random Forest, XGBoost) and neural networks, are often considered “black-box” models due to the complexity of their decision-making process, which makes it difficult to interpret their predictions intuitively. While these models tend to offer higher predictive accuracy, their complexity limits understanding. However, explainability methods can be applied to provide insights into their reasoning [28].

## 4. Methodology

Figure 1 presents the proposed framework of our study which is structured into several stages. First, the raw data undergo a data preprocessing phase, where essential cleaning and transformation steps are applied. Ten supervised classification algorithms were trained with hyperparameter optimization integrated into the training process, and subsequently evaluated using appropriate performance metrics. The predictive models employed include Support Vector Classifier (SVC) with linear, RBF, and polynomial kernels, Logistic Regression, Decision Tree, Random Forest, Extreme Gradient Boosting (XGBoost), k-Nearest Neighbors (KNN), Naive Bayes, and Multilayer Perceptron (MLP). The models were selected as they cover a range of algorithmic families—including linear models, tree-based methods, kernel-based classifiers, and neural networks—enabling a balanced comparison in terms of complexity, interpretability, and performance. As an additional analysis, a feature selection process was carried out to identify the most relevant variables and evaluate their effect on model outcomes. Finally, to enhance transparency, explainability techniques are applied, providing insights into model decision making. The proposed system has been designed to reliably predict metastasis in PPGL patients, aiding in early detection and informed clinical choices.

### 4.1. Dataset

The dataset used in this study was collected from Pamporaki et al. [4]. It includes information on patients with and without metastatic PPGLs, obtained from seven specialized hospitals in Germany, the Netherlands, Poland, and the United States, and comprises a total of 788 patient samples. Of these, 223 (28.3%) corresponded to metastatic cases and 565 (71.7%) to non-metastatic cases. As reported in the original publication, clinical protocols were approved by the respective local ethics committees.

This dataset contained the following variables: Patient ID, Sex, Age, Plasma normetanephrine (NMN), Plasma metanephrine (MN), Plasma methoxytyramine (MTY), Previous History, Location, Presence of SDHB, Tumor Category, Spherical Volume, and Metastatic, indicating the presence or absence of metastasis and serving as the target variable for model prediction. Additionally, an Age Group variable was introduced to categorize patients as Pediatric (0–18 years) or Adult (≥18 years), based on the 18-year threshold widely accepted across most European countries. The full description of the dataset is shown in Table 1.

Plasma concentrations of NMN, MN, and MTY were included in the dataset due to their biological relevance in PPGLs [41]. These compounds are metabolites of catecholamines, which play a key role in the function of these tumors: Metanephrine (MN) is derived from adrenaline (epinephrine), normetanephrine (NMN) from noradrenaline (norepinephrine), and methoxytyramine (MTY) from dopamine. Since PPGLs arise from cells that produce catecholamines, their metabolic byproducts can act as biochemical markers, offering insight into tumor behavior and disease progression. Notably, elevated plasma MTY and dopamine levels are strongly associated with metastasis [42] independently of tumor size, suggesting a direct link to malignant transformation and correlating with SDHB mutations and extra-adrenal tumor location. In contrast, increases in NMN and norepinephrine in metastatic tumors appear more related to tumor burden, with norepinephrine also known to enhance cancer cell proliferation. Conversely, MN levels tend to be lower in metastatic patients, especially in extra-adrenal tumors that typically lack epinephrine secretion [25]. These metabolic changes reflect the tumor microenvironment, where catecholamines and their metabolites influence carcinogenesis and tumor progression [43].

Biomedical sensors have been used to obtain plasma concentrations of NMN, MN, and MTY. Specifically, liquid chromatography with tandem mass spectrometry (LC-MS/MS) and HPLC with electrochemical detection (LC-ECD) were used to perform the biochemical tests.

### 4.2. Data Preprocessing

The data preprocessing phase is fundamental for the creation of machine learning models. This phase includes loading, cleaning, and transforming the dataset to ensure data quality and compatibility with subsequent predictive modeling and analytical procedures.

**Class Imbalance Handling:** In order to address the class imbalance problem, the Random OverSampling technique was applied. This method increases the representation of the minority class by replicating its instances, thereby promoting a more balanced class distribution and improving the reliability of model predictions.**Missing Value Treatment:** Missing values were identified, revealing a small number of omissions in the dataset. To preserve data integrity, imputation was applied, using the mean value for missing entries in the numeric variables Age, Plasma NMN, Plasma MN, Plasma MTY, and Spherical volume.**Outlier Detection:** Outlier detection was performed using box plot analysis for numeric variables. An extreme value was identified in the Plasma NMN variable, with a concentration over twice as high as the next largest measurement. Due to its significant deviation from the central tendency, it is likely a result of measurement error. Consequently, this outlier was removed to maintain the quality of the data. Figure 2 presents the boxplots of the three plasma metabolites on a logarithmic scale to better visualize and compare their distributions.**Encoding Categorical Variables:** Categorical columns were transformed into a numerical format using one-hot encoding, where each unique category was represented as a separate binary column. This transformation avoids introducing ordinal relationships between categories and ensures compatibility with machine learning algorithms.

### 4.3. Optimization Procedure and Evaluation Metrics

A nested cross-validation strategy was implemented to ensure a reliable evaluation of model performance and avoid overfitting during hyperparameter tuning. First, a 5-fold cross-validation was used to tune hyperparameters using GridSearchCV, applied only on the training data. Then, a 10-fold cross-validation was performed, where, in each iteration, the model—trained with the best hyperparameters—was evaluated on a separate test set to assess its generalization performance.

To standardize all input variables prior to model training, three different data scaling techniques were applied: StandardScaler, which standardizes features by removing the mean and scaling to unit variance; MinMaxScaler, which transforms features to a defined range (default 0 to 1); and RobustScaler, which scales data based on the interquartile range. The hyperparameter tuning process identified the optimal combination of model parameters and scaling method for each model.

The performance of metastasis prediction was evaluated based on Accuracy, Precision, Recall, F1-Score, and AUC.

**Accuracy** measures the proportion of correctly classified instances among all samples.**Precision** measures the proportion of correctly predicted positive instances out of all instances predicted as positive, indicating the model’s ability to avoid false positives.**Recall** (also known as sensitivity) measures the proportion of correctly identified positive instances among all actual positive cases, reflecting the model’s ability to capture all true positives.**F1-Score** represents the harmonic mean between Precision and Recall, offering a single metric that balances both concerns.**AUC (Area Under the Curve)** quantifies the overall ability of the model to discriminate between positive and negative classes across all classification thresholds; a higher AUC indicates better overall model performance.

## 5. Results

This section presents the results obtained throughout the model development and evaluation phases. First, we assess the performance of the algorithms when trained with all available features. Subsequently, we conduct a feature selection analysis to identify the most relevant variables, and then re-evaluate the models using only those key features. Finally, we explore the explainability of the best-performing model through global and local interpretability techniques.

### 5.1. Evaluation of the Models

The results produced by each model using all the features available in our dataset are presented in Table 2.

As observed, among the models trained with the complete feature set, the Random Forest algorithm consistently achieved the highest performance across all evaluation metrics on average, obtaining an accuracy of 96.3%, a precision of 96.5%, a recall of 96.3%, an F1-score of 96.3%, and an AUC of 0.963. These results, evaluated on the held-out test sets of the outer cross-validation, indicate that Random Forest not only made accurate predictions but also maintained a strong balance between identifying true metastatic cases and minimizing false positives.

Following closely, the XGBoost model achieved competitive performance, with an accuracy of 95.3%, a precision of 95.6%, a recall and F1-score of 95.3%, and an AUC of 0.954. The consistently strong results of tree-based methods, particularly Random Forest and XGBoost, underscore their ability to capture relevant patterns in the dataset. Among the remaining models, Decision Tree and SVC with RBF kernel showed solid performance, with scores above 90% across all evaluated metrics. In contrast, the relatively lower performance of linear models such as Logistic Regression and SVC linear, may indicate the presence of non-linear relationships between features. Bernoulli Naive Bayes presented the lowest performance. However, it still achieved an accuracy of 82.9%, suggesting some predictive capacity, though it was clearly limited in capturing the patterns present in the data.

All models showed statistically significant differences in performance when compared to the Random Forest classifier (p<0.05 or p<0.01), based on paired *t*-tests, highlighting its superior predictive ability.

When comparing our results with those reported in the baseline study by Pamporaki et al. [4], we observe significant improvements in performance. It is important to note that while we used the same dataset of ten features as in their work, we introduced an additional categorical variable derived from age, which does not compromise the validity of the comparison.

**Decision Tree Classifier:** The baseline study reported an accuracy of 91.4%, an F1-score of 75.0%, and an AUC of 0.893 for the Decision Tree classifier. In contrast, our Decision Tree model achieved an accuracy of 93.5%, an F1-score of 93.5%, and an AUC of 0.935. These results represent a 2.1% increase in accuracy, a 18.5% improvement in F1-score, and a 0.042 increase in AUC.**Support Vector Classifier (SVC):** The linear SVC model in previous work achieved an accuracy of 83.4%, an F1-score of 65.1%, and an AUC of 0.924. In our study, we tested three different SVC kernels: linear, polynomial, and RBF. The SVC with the RBF kernel outperformed the others, achieving an accuracy of 91.9%, an F1-score of 91.9%, and an AUC of 0.919. This represents an improvement of 8.5% in accuracy, a 26.8% increase in F1-score, and a slight decrease of 0.005 in AUC.**Best Model Comparison. AdaBoost vs. Random Forest:** The best model in the baseline study was the AdaBoost ensemble, which achieved an accuracy of 85.6%, an F1-score of 67.2%, and an AUC of 0.940. Our best model, the Random Forest classifier, outperformed this with an accuracy of 96.3%, an F1-score of 96.3%, and an AUC of 0.963. These results represent a 10.7% improvement in accuracy, a 29.1% increase in F1-score, and a 0.023 improvement in AUC.

Overall, the Random Forest model emerged as the most robust performer in our study, surpassing other classifiers in key metrics. This reinforces the idea that ensemble methods are particularly well-suited for this task, effectively handling diverse data patterns and providing reliable predictions.

All model development and evaluation was conducted using an Intel Xeon CPU (two virtual cores) with 13 GB of RAM, which provides a moderately powerful baseline. This supports the potential viability of deploying the model even in resource-limited settings with basic computational capabilities such as low-power CPUs or embedded systems, assuming minimal preprocessing requirements. However, further investigation is needed before integrating such models into real-world clinical practice, including careful evaluation of local infrastructure, power availability, data acquisition processes, and device compatibility.

Beyond predictive performance, we assessed whether Random Forest models could be feasibly used in real-time clinical settings, considering their computational demands and resource requirements. Since models are trained offline and deployed in pre-trained form, training time was not considered a limiting factor. Inference time was measured by repeatedly predicting a single representative sample 100 times and computing the average, finding that the Random Forest model required approximately 235 ms per sample, with a model size of around 1.14 MB in RAM and 16 MB on disk. These metrics suggest that real-time prediction is technically feasible on standard computing devices used in healthcare environments.

### 5.2. Feature Selection

Feature selection was conducted to identify the most relevant variables and reduce unnecessary noise in the data. The main goal of this step goes beyond improving algorithm performance, it also aims to reduce the amount of information required from patients. By focusing on the most informative features, the approach helps reduce the invasiveness of data collection, save time, and lower resource demands, ultimately contributing to a more efficient and less demanding diagnostic process.

In the feature selection process, three different methods were employed: Chi-square tests, Mutual Information, and Information Gain. These techniques were chosen for their complementary capabilities in identifying both linear and non-linear relationships between features and the target variable.

The Chi-square test was applied to assess the association between categorical variables and the target variable, revealing key features that were most significantly related to the outcome. Mutual Information quantifies the amount of shared information between a feature and the target variable. A high mutual information score suggests that the feature carries strong predictive relevance, capturing both linear and non-linear dependencies. Information Gain is based on entropy, which measures the uncertainty of the data. It quantifies how much a feature reduces this uncertainty when predicting the target variable. A higher Information Gain indicates that the feature provides more relevant information for predicting the outcome.

The results of these statistical feature selection techniques are summarized in Table 3, which presents the normalized scores for Chi-Square, Mutual Information, and Information Gain across the features of the dataset.

From the results obtained through these methods, we identified the most relevant variables by focusing on those that consistently achieved high scores across all three evaluated techniques. These included Previous history, Presence of SDHB, and Location. Their strong and stable association with the target variable across the three different statistical approaches clearly demonstrates their potential to serve as highly relevant predictors in this context.

To provide further insight into these findings, we have included additional biological context regarding the role of SDHB mutations in promoting metastasis. The SDHB gene encodes a key subunit of mitochondrial complex II (succinate dehydrogenase), which plays an essential role in helping mitochondria produce energy through two related processes: the TCA cycle and the electron transport chain. Both pathways require oxygen and generate ATP, the main energy source for cells. When SDHB is mutated, mitochondria cannot carry out normal energy production using oxygen, and cells shift towards anaerobic glycolysis, a pathway that does not require oxygen to meet their energy needs, even when oxygen is available. This phenomenon, known as the Warburg effect, is commonly seen in cancer cells. It leads to increased glucose uptake and the accumulation of molecules such as succinate and reactive oxygen species (ROS), which can damage cellular structures and promote tumor growth. At the same time, impaired mitochondrial function may weaken apoptosis, allowing damaged cells to evade death and potentially metastasize [44].

#### 5.2.1. Study with the Three Most Important Features

Following the initial evaluation using all available features, a second analysis was carried out using only the three variables identified as the most relevant: Previous history, Presence of SDHB, and Location.

The results obtained for the different models, using the configuration with the most significant features are presented in Table 4.

Based on the evaluation metrics, the majority of the models exhibited very similar performance when trained exclusively on the three most relevant features. Accuracy, precision, recall, F1-score all ranged closely between approximately 82% and 84%, suggesting that these features encapsulate a significant portion of the predictive information in the dataset. This uniformity across models indicates that, despite differences in algorithmic approach, most classifiers were able to extract comparable insights from the reduced feature space. The only clear exception was the KNN model, which showed considerably lower scores across all metrics, likely due to its dependence on distance calculations, which are less effective in low-dimensional spaces dominated by categorical variables.

#### 5.2.2. Comparison of Results

We further compared the results obtained in both previous studies, analyzing how the different approaches and features employed influenced the prediction of the metastatic condition of cancer, as depicted in Figure 3.

When comparing model performance using all features versus only the top three selected, we observed a consistent drop in accuracy. For instance, Random Forest decreased from 96.3% to 83.5%, and XGBoost from 95.3% to 82.7%. This trend confirms the benefit of using the full feature set to maximize predictive accuracy.

Interestingly, some models, such as Logistic Regression, BernoulliNB, MLP, and SVC linear, maintained similar performance in both studies, suggesting that their performance is less influenced by the number of features used in training. In contrast, KNN was notably affected, dropping from 86.9% to 64.7%, highlighting its reliance on more input variables.

All models, with the exception of MLP, Logistic Regression, linear SVC, and BernoulliNB, demonstrated statistically significant differences in performance when comparing the results obtained using all features versus only the top three features. This was confirmed by paired *t*-tests, with significance levels of p<0.01.

Despite the general reduction in accuracy, the models trained with only three features still produced competitive results—particularly notable given the substantial reduction in input dimensionality. These findings highlight the trade-off between model complexity and predictive performance: while using all features yields optimal accuracy, a small, well-selected subset can still provide strong predictive power, supporting more practical, efficient, and potentially less invasive clinical implementations.

### 5.3. Explainability of the Random Forest Model

Individual decision trees are inherently interpretable due to their transparent structure. However, Random Forests, as ensembles of multiple decision trees, lose this direct interpretability, since the combined effect of numerous trees makes it difficult to identify the precise reasoning behind a given prediction. This section focuses on the explainability of the Random Forest model, the best-performing in our evaluation, to gain a deeper understanding of its decision-making process. To achieve this, both global and local explainability techniques were applied. Global methods quantify the contribution of each feature to the model’s overall behavior, while local approaches explain predictions for specific cases, like those of individual patients.

#### 5.3.1. Global Explainability

In order to better understand the model’s predictions globally, we used SHAP and Feature Importance.

As mentioned before, SHAP not only provides local interpretability but also explains model behavior on a global scale by expressing its predictions as a combination of feature contributions. In other words, it quantifies the impact of each feature on the overall behavior of the model. Figure 4 illustrates the SHAP summary plot for the Random Forest model. Each dot represents a SHAP value for a feature in a specific instance. The x-axis corresponds to the SHAP values, which quantify the impact of a feature, while the y-axis lists the features ranked by importance. The color gradient indicates the feature values, with red representing higher values and blue representing lower values. Positive SHAP values indicate a contribution towards the condition of having metastasis, while negative SHAP values indicate a contribution towards the absence of metastasis.

At a global level, Previous history appears as the most influential feature in the model. This categorical variable, encoded as Previous history_YES and Previous history_NO, indicates whether the patient has a prior history of cancer. As shown in the plot, high values of Previous history_YES (red points) are associated with positive SHAP values, meaning that a history of cancer strongly contributes to an increased predicted probability of metastasis. Conversely, Previous history_NO is associated with negative SHAP values, decreasing the predicted likelihood of metastasis.

Another important feature is Spherical volume. Higher tumor volumes (red points) are associated with positive SHAP values, indicating a greater probability of metastatic spread. This is consistent with existing literature, where tumors larger than approximately 4 cm in diameter (equivalent to 33.5 cm^3^ in spherical volume) are widely recognized as indicative of increased tumor aggressiveness and metastatic potential. According to an international expert consensus on the management of PPGLs [45], tumors smaller than 4 cm are generally suitable for minimally invasive surgical approaches (e.g., laparoscopic), which offer faster recovery and less morbidity. Conversely, tumors larger than 6 cm are recommended for open surgery due to higher risks of local invasion, recurrence, and metastatic spread, while those between 4 and 6 cm require a personalized surgical approach based on individual tumor characteristics.

Regarding tumor location, extra-adrenal tumors are linked to elevated metastatic potential, while adrenal locations tend to contribute negatively to the prediction. Plasma biomarkers such as MN, NMN, and MTY also contribute to the model. Metastatic PPGLs are generally characterized by elevated MTY, reduced MN, and consistently high NMN, a pattern that is reflected in their respective SHAP values.

The Presence of SDHB mutation is relevant as well: patients with confirmed SDHB mutations (Presence of SDHB_YES) tend to have higher SHAP values, suggesting increased metastasis risk. This aligns with international expert consensus [45], which identifies germline SDHB mutations as a key factor in risk stratification due to their strong association with aggressive tumor behavior and a lifetime metastatic risk of approximately 35–40%. These tumors are frequently locally invasive, prone to recurrence, and may present atypical biochemical profiles, which complicate management. Consequently, current guidelines recommend early and aggressive surgery regardless of size, along with lifelong follow-up.

Finally, to a lesser extent, variables such as Age, Tumor category, and Sex also contribute to the model, though with varying degrees of influence. For example, slightly higher age shows a weak association with increased metastatic risk, while tumor subtypes and sex appear to have limited predictive value in this model.

On the other hand, Feature Importance is a technique used to assess the contribution of each feature in a predictive model by measuring the decrease in impurity when a feature is used to split data within a decision tree. In Random Forest, this is typically quantified as Gini importance, where higher values indicate greater influence on the model’s predictions.

Figure 5 presents the feature importance ranking for the Random Forest model. The results show that Spherical volume and Previous history are the most important predictors of metastasis, with both contributing significantly to the model’s classification. In addition, Plasma MN and Plasma MTY also exhibit considerable influence on the prediction outcome, highlighting the role of biochemical markers in the classification task. These findings closely align with the SHAP analysis.

#### 5.3.2. Local Explainability

To gain a deeper understanding of the model’s predictions on a local level, we employed SHAP.

Figure 6a illustrates a SHAP waterfall plot for a patient exhibiting a high probability of metastasis (f(x)=0.98). The baseline prediction (E[f(X)]=0.501) is modified by each feature’s contribution. Positive SHAP values (red) increase the likelihood of metastasis, whereas negative values (blue) reduce it. In this case, the strongest contributors to the high score are the patient’s Previous history of disease (+0.28 when combining YES and NO indicators) and a large tumor volume (+0.11), with a size of 445.6 cm^3^, which exceeds the threshold typically associated with increased metastatic risk. The presence of an SDHB mutation (+0.04) further raises the prediction, consistent with its known link to aggressive tumor behavior. Additional positive contributions come from elevated plasma concentrations of MTY and MN, and low concentrations of MN. In contrast, tumor location in the adrenal gland exerts a negative effect (−0.04 each), reflecting its lower metastatic potential compared to extra-adrenal tumors.

To further explore individual predictions, we also examine a SHAP waterfall plot for a patient with a lower metastasis probability (f(x)=0.205), as shown in Figure 6b. In this case, the Spherical volume is the main factor lowering the prediction (−0.23), with a size of 1.9 cm^3^, below the threshold typically associated with larger, more aggressive tumors. The absence of Previous disease history also lowers the score (−0.11 when combining YES and NO indicators), as does the presence of low plasma concentrations of NMN and MTY. On the other hand, tumor location increases the predicted risk (+0.08 each), as the tumor is extra-adrenal, a type more commonly linked to metastatic behavior.

These findings illustrate that the Random Forest model predominantly bases its predictions on well-established clinical indicators of metastatic risk. Specifically, having a previous history of PPGLs, presenting a large tumor volume, elevated levels of MTY and NMN, extra-adrenal tumor location, and the presence of SDHB mutations all contribute to higher predicted probabilities of metastasis. These insights, derived from both global and local analyses, align with patterns observed in clinical practice and previously reported studies.

## 6. Conclusions and Future Work

In this study, we developed a machine learning-based approach to predict metastasis in paragangliomas and pheochromocytomas (PPGLs), incorporating explainability techniques to enhance model transparency and trust. Our findings highlight the effectiveness of the Random Forest algorithm, which achieved over 96% performance when using all available features. Additionally, while using the full feature set yielded the best predictive results, models trained on only three variables still performed competitively. This highlights a relevant trade-off between model complexity and predictive accuracy: although reduced models are less accurate, they may offer more practical, and potentially less invasive solutions in real-world clinical scenarios.

The implementation of explainability methods allowed us to interpret model predictions and identify the most relevant factors for metastasis detection. These results emphasize the importance of explainability in clinical applications, as they provide valuable insights for medical decision-making and facilitate the adoption of machine learning models in healthcare settings.

Despite these advancements, several challenges remain to be addressed in future research. Validating the model with external datasets from diverse populations and clinical settings will be essential to assess its reliability across different contexts. Furthermore, integrating multimodal data, including genetic information and medical imaging, could further enhance the model’s predictive capabilities and its practical use in clinical settings. Also, given the relatively small model size and low inference time observed for the Random Forest classifier, future research could explore its deployment in low-resource environments, such as embedded systems or edge computing devices. These platforms are increasingly used in clinical settings to enable real-time decision support close to the patient. Evaluating the feasibility of implementing lightweight models on such devices could enhance accessibility and broaden the impact of predictive tools, especially in resource-constrained or decentralized healthcare contexts. Finally, incorporating complementary explainability techniques, such as LIME, could enrich the interpretation of the model’s behavior and support more comprehensive, trustworthy decision-making.

In conclusion, our work highlights the potential of machine learning in predicting metastasis in PPGLs and underscores the critical role of explainability in clinical implementation. Future research should focus on optimizing and validating these models to maximize their impact in medical practice.

## Figures and Tables

**Figure 1 sensors-25-04184-f001:**
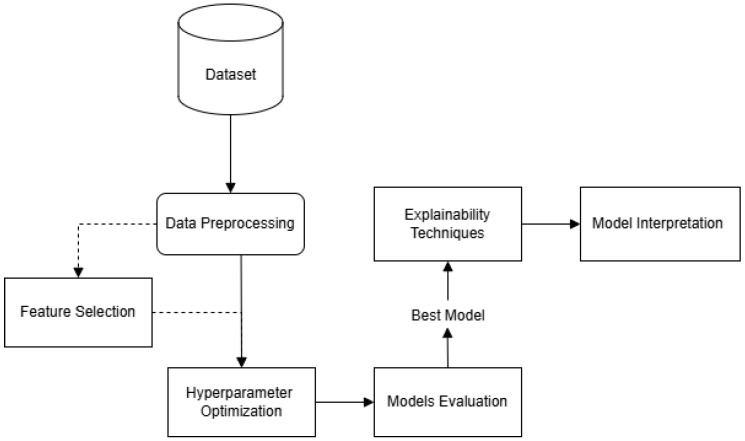
Overview of the proposed methodology.

**Figure 2 sensors-25-04184-f002:**
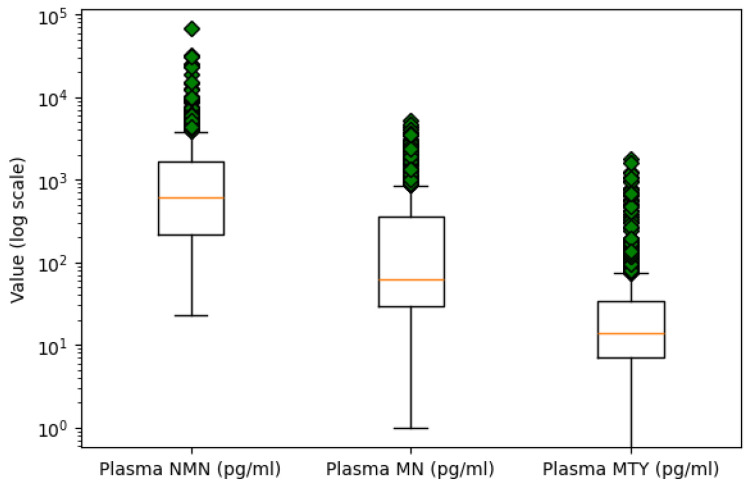
Outlier analysis of plasma metabolites using logarithmic scale boxplots.

**Figure 3 sensors-25-04184-f003:**
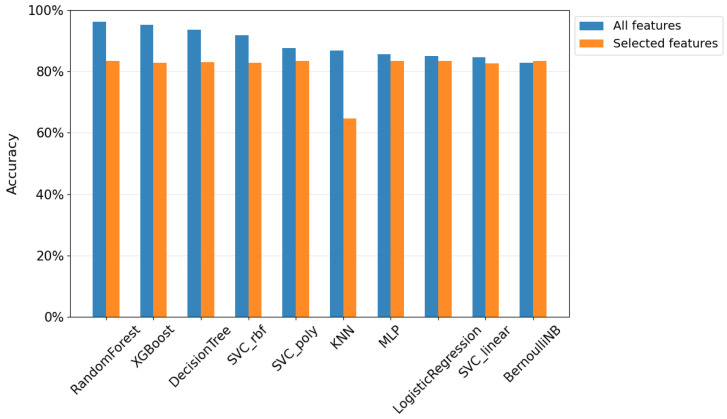
Impact of feature selection on classification accuracy.

**Figure 4 sensors-25-04184-f004:**
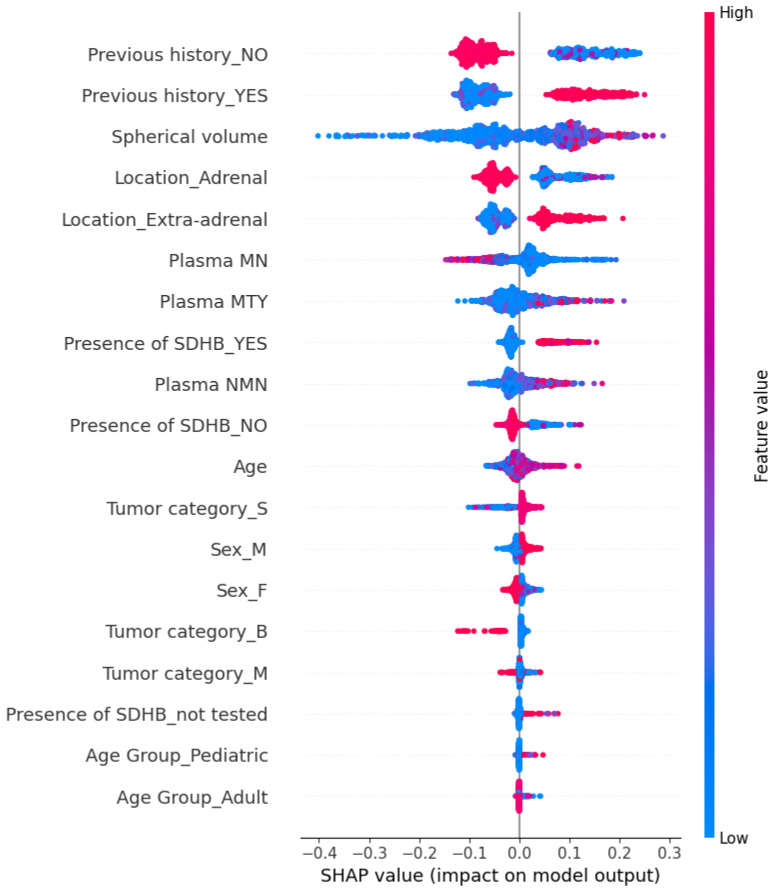
SHAP summary plot for Random Forest predictions.

**Figure 5 sensors-25-04184-f005:**
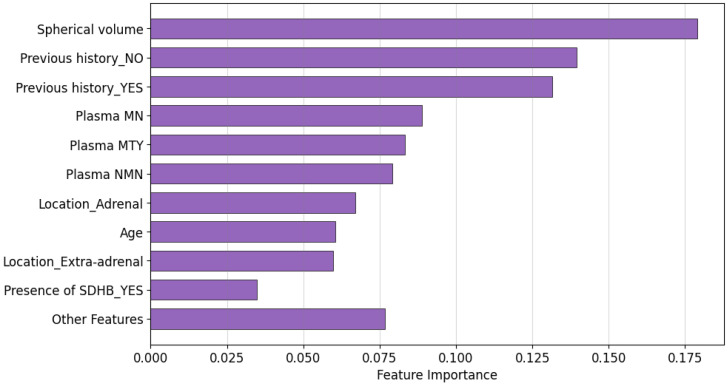
Feature importance ranking for the Random Forest model.

**Figure 6 sensors-25-04184-f006:**
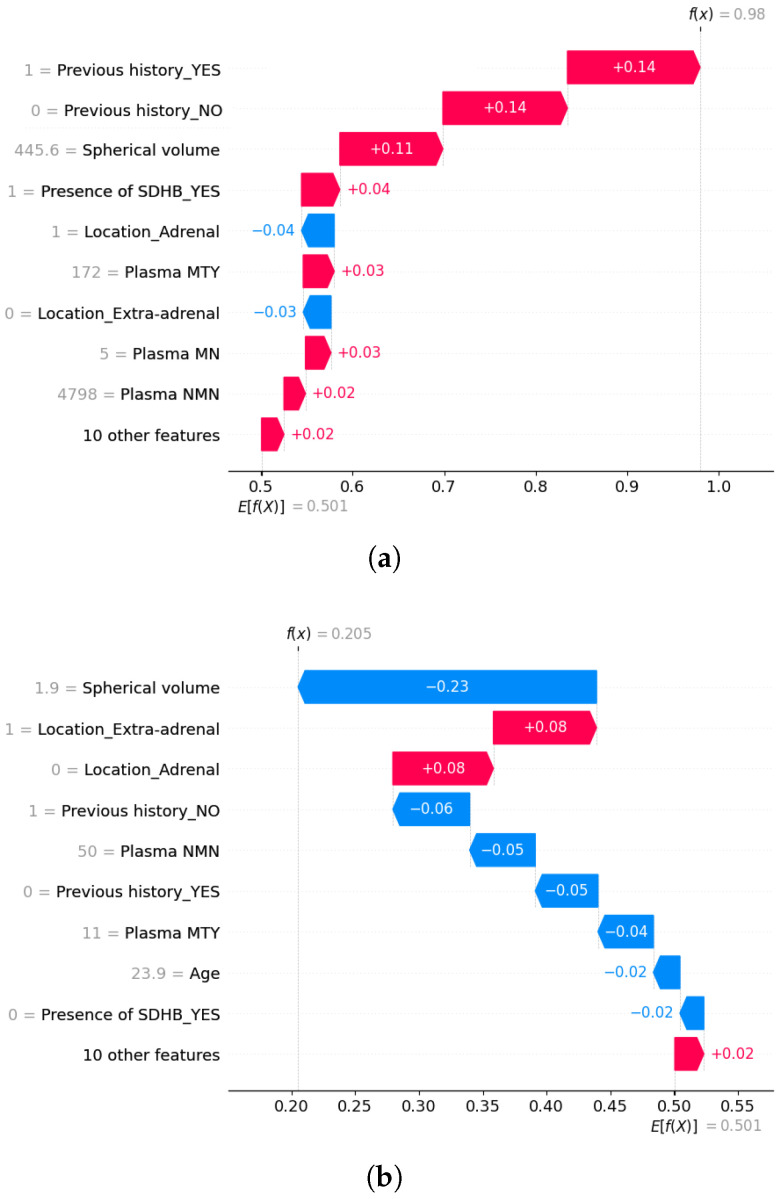
SHAP waterfall plots for two different patients. (**a**) Patient with high metastasis risk. (**b**) Patient with low metastasis risk.

**Table 1 sensors-25-04184-t001:** Description of the dataset features.

Feature Name	Type	Description
Patient ID	Integer	Unique identifier assigned to each patient
Sex	Categorical	Patient’s gender (Male/Female)
Age	Float	Age of the patient at the time of first tumor diagnosis (in years)
Plasma NMN	Float	Concentration of Normetanephrine (NMN) in plasma (pg/mL)
Plasma MN	Float	Concentration of Metanephrine (MN) in plasma (pg/mL)
Plasma MTY	Float	Concentration of Methoxytyramine (MTY) in plasma (pg/mL)
Previous history	Categorical	Indicates whether the patient had previous PPGL diagnoses (Yes/No)
Location	Categorical	Tumor location: adrenal or extra-adrenal
Presence of SDHB	Categorical	Indicates presence of SDHB gene mutation (Positive/Negative/Not tested)
Tumor category	Categorical	Tumor type: Solitary (S), Bilateral (B), or Multifocal (M)
Spherical volume	Float	Estimated spherical volume of the tumor (cm^3^)
Metastatic	Categorical	Indicates whether metastasis is present (target variable) (Yes/No)

**Table 2 sensors-25-04184-t002:** Performance of the models with all available features. Values are reported as mean ± standard deviation. Asterisks indicate levels of statistical significance: * p<0.05, ** p<0.01.

Model	Accuracy		Precision		Recall		F1-Score		AUC	
	Mean	Std	Mean	Std	Mean	Std	Mean	Std	Mean	Std
RandomForest	0.963	0.014	**0.965**	0.013	0.963	0.014	0.963	0.014	0.963	0.014
XGBoost	0.953 *	0.016	**0.956** *	0.014	0.953 *	0.016	0.953 *	0.016	0.954 *	0.016
DecisionTree	0.935 **	0.025	**0.940** **	0.022	0.935 **	0.025	0.935 **	0.026	0.935 **	0.027
SVC_rbf	0.919 **	0.031	**0.921** **	0.031	0.919 **	0.031	0.919 **	0.031	0.919 **	0.031
SVC_poly	0.876 **	0.032	**0.877** **	0.032	0.876 **	0.032	0.876 **	0.032	0.876 **	0.033
KNN	0.869 **	0.043	**0.871** **	0.042	0.869 **	0.043	0.869 **	0.043	0.868 **	0.043
MLP	0.856 **	0.024	**0.857** **	0.024	0.856 **	0.024	0.855 **	0.024	0.853 **	0.028
LogisticRegression	0.850 **	0.029	**0.852** **	0.030	0.850 **	0.029	0.850 **	0.029	0.849 **	0.031
SVC_linear	0.847 **	0.018	**0.849** **	0.017	0.847 **	0.018	0.847 **	0.018	0.846 **	0.018
BernoulliNB	0.829 **	0.032	**0.830** **	0.032	0.829 **	0.032	0.829 **	0.032	0.827 **	0.034

**Table 3 sensors-25-04184-t003:** Feature selection scores for Chi-Square, Mutual Information, and Information Gain.

Feature	Chi-Square	Mutual Information	Information Gain
Previous history	1.000	1.000	1.000
Presence of SDHB	0.739	0.563	0.574
Location	0.587	0.607	0.633
Tumor category	0.086	0.014	0.084
Spherical volume	0.072	0.586	0.251
Plasma MTY	0.069	0.594	0.126
Age Group	0.024	0.000	0.000
Sex	0.021	0.026	0.020
Plasma MN	0.015	0.243	0.093
Plasma NMN	0.010	0.051	0.073
Age	0.000	0.117	0.100

**Table 4 sensors-25-04184-t004:** Performance of the models with the three most relevant features. Values are reported as mean ± standard deviation. Asterisks indicate levels of statistical significance: ** p<0.01.

Model	Accuracy		Precision		Recall		F1-Score		AUC	
	Mean	Std	Mean	Std	Mean	Std	Mean	Std	Mean	Std
RandomForest	0.835	0.030	**0.844**	0.032	0.835	0.030	0.834	0.031	0.834	0.033
BernoulliNB	0.835	0.030	**0.836**	0.030	0.835	0.030	0.835	0.030	0.834	0.031
LogisticRegression	0.835	0.030	**0.836**	0.030	0.835	0.030	0.835	0.030	0.834	0.031
SVC_poly	0.835	0.030	**0.836**	0.030	0.835	0.030	0.835	0.030	0.834	0.031
MLP	**0.835**	0.031	**0.835**	0.031	**0.835**	0.031	0.834	0.031	0.833	0.032
DecisionTree	0.830	0.023	**0.838**	0.023	0.830	0.023	0.829	0.023	0.829	0.024
SVC_rbf	0.829	0.034	**0.839**	0.036	0.829	0.034	0.829	0.034	0.831	0.036
XGBoost	0.827	0.023	**0.835**	0.024	0.827	0.023	0.827	0.024	0.828	0.024
SVC_linear	0.826	0.035	**0.828**	0.034	0.826	0.035	0.826	0.035	0.824	0.036
KNN	0.647 **	0.053	**0.780** **	0.033	0.647 **	0.053	0.600 **	0.059	0.647 **	0.029

## Data Availability

The original data presented in the study are openly available on the online platform Zenodo at https://doi.org/10.5281/zenodo.7749613 (accessed on 21 January 2024).

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
