# Peer review of "Prediction of Metastasis in Paragangliomas and Pheochromocytomas Using Machine Learning Models: Explainability Challenges"

_sensors, 2025, doi:10.3390/s25134184_

Round 1
Reviewer 1 Report
Comments and Suggestions for Authors
In this paper, the Authors address the problem of predicting the occurrence of metastasis related to paragangliomas and pheochromocytoma, from clinical data of various nature, by employing Machine Learning algorithms. In particular, the Authors analyzed the issue also from the point of view of model explainability, a crucial characteristic to ensure the growth of a trustworthy
Artificial Intelligence. The article’s content is highly readable, and the results are well-presented. For this reason, my feedback about the paper is positive. In the following, I’ve reported some minor suggestions to further improve the manuscript’s readability:
1. Row 188: I suggest inserting a bibliographic reference to support the statement about the relevance of PPGLs.
2. The section Results includes also the description of the characteristics of the ML techniques. I believe that organizing the information about the ML models, the description of the optimization procedure, and the techniques/parameters adopted to assess the models’ performances in a
separate section, e.g., a subsection of section 4, could further enhance the readability.
3. Table 2: Did the Authors try to determine the dispersion, e.g., the standard deviation on the folds of CV, associated with the performances of the ML models? This could help in monitoring the level of overfitting.
4. The Authors reported the accuracy within the text as a percentage and all the other parameters (precision, recall, etc.) as a fraction. For the sake of clarity, I suggest adopting the same way throughout the manuscript.
5. subsection 5.2.1: How did the Authors conclude that the difference between the performance parameters with the three most important features and the same parameters determined with all the features is not significant? Again, I believe that determining the standard deviations or analogous parameters representative of the dispersion could help in establishing if the differences between the performance parameters are significant.
Reviewer 2 Report
Comments and Suggestions for Authors
In this work, the authors propose an architecture that combines data mining, machine learning, and explainability techniques to improve predictions of metastatic disease in these types of cancer and enhance trust in the models. However, there are still some problems should be solved before next step. For example:
- Although the paper includes biomarkers such as plasma metabolites (NMN, MN, MTY) and SDHB mutations, it does not delve into the biological association between these characteristics and metastasis mechanisms. For example, the pathological mechanism of elevated MTY concentration and tumor invasiveness has not been analyzed, or how SDHB mutations promote metastasis through mitochondrial dysfunction.
- Only mentioning the use of random oversampling to handle class imbalance, without specifying the standardization method for plasma metabolite concentration (such as whether Z-score or logarithmic transformation is used), and without evaluating the impact of outliers (such as plasma NMN outliers) on the model.
- Using only clinical and biochemical data without integrating imaging features (such as tumor MRI texture) or genomic data (such as whole exome sequencing mutation profiles) limits the model's ability to capture tumor heterogeneity.
- The inference time, computational resource requirements, or sample size requirements of the model have not been evaluated, and there is a lack of analysis on the feasibility of bedside detection. For example, the high complexity of the Random Forest model may limit its application in resource limited environments.
- Only mentioning that the data is anonymized without providing the approval number of the ethics review committee, nor specifying the specific process of patient informed consent, does not comply with the ethical norms of biomedical research.
- Although SHAP analysis identified previous history and tumor volume as key features, it did not explain its threshold in conjunction with clinical guidelines (such as the clinical significance of tumor volume>33.5 cm ³), nor did it explain how the SHAP value of SDHB mutations corresponds to existing risk stratification criteria.
Reviewer 3 Report
Comments and Suggestions for Authors
I would like to thank the authors for submitting this manuscript and congratulate them on a well-structured, technically competent, and clearly written study. The topic addressed is highly relevant, as predicting metastatic potential in paragangliomas and pheochromocytomas remains a clinical challenge. The integration of machine learning techniques with explainability mechanisms is timely and valuable for improving both predictive accuracy and trust in model output - particularly in sensitive applications such as oncology.
I found the methodology to be well explained, and the results accessible to readers from both medical and technical backgrounds. The manuscript reflects a solid command of both domains and demonstrates the authors’ thoughtful approach to data handling, feature selection, and interpretability of results. I believe this work will be of interest to a multidisciplinary audience and can support the integration of AI into future clinical decision-making frameworks.
That said, I have a few minor comments and suggestions:
First, the Abstract could benefit from the inclusion of specific numerical results, such as the performance metrics (e.g., accuracy, AUC, precision, recall) of the best-performing model. This would help convey the strength of the findings upfront and make the abstract more informative for readers scanning the content.
Second, while the Introduction provides adequate background and motivation, the main objective of the study is not clearly and explicitly stated. Even though the goal can be inferred from the context, a clearer statement of the research aim would help orient the reader more effectively.
Third, there are a few minor formatting issues throughout the text. For instance, on line 105 the acronym ROC is not explained. Similarly, the abbreviation KNN is introduced without definition on line 113 and only explained on line 149. The same applies to NMN, MN, and MTY.
Finally, in Figure 2, the boxplots for some features are nearly invisible due to scale differences. I suggest considering a logarithmic scale for the y-axis, or alternative plotting strategies, to enhance visual clarity.
These remarks are of a recommendatory nature and do not diminish the overall quality of the work. I commend the authors for a high-level, interdisciplinary contribution and would recommend acceptance after minor revision.
Round 2
Reviewer 2 Report
Comments and Suggestions for Authors
The revised manuscript can be accepted in this version.